# Metabolomic Analysis Evidences That Uterine Epithelial Cells Enhance Blastocyst Development in a Microfluidic Device

**DOI:** 10.3390/cells10051194

**Published:** 2021-05-13

**Authors:** Vanessa Mancini, Alexandra C. Schrimpe-Rutledge, Simona G. Codreanu, Stacy D. Sherrod, John A. McLean, Helen M. Picton, Virginia Pensabene

**Affiliations:** 1School of Electronic and Electrical Engineering, University of Leeds, Leeds LS2 9JT, UK; elvm@leeds.ac.uk; 2Center for Innovative Technology (CIT), Department of Chemistry, Vanderbilt University, 7300 Stevenson Center Lane, Nashville, TN 37235, USA; alexandra.c.rutledge@vanderbilt.edu (A.C.S.-R.); simona.codreanu@vanderbilt.edu (S.G.C.); stacy.d.sherrod@vanderbilt.edu (S.D.S.); john.a.mclean@Vanderbilt.edu (J.A.M.); 3Reproduction and Early Development Research Group, Discovery and Translational Science Department, Leeds Institute of Cardiovascular and Metabolic Medicine, School of Medicine, University of Leeds, Leeds LS2 9JT, UK; H.M.Picton@leeds.ac.uk; 4Leeds Institute of Medical Research, University of Leeds, Leeds LS2 9JT, UK

**Keywords:** embryo culture, microfluidics, metabolomics

## Abstract

Here we report the use of a microfluidic system to assess the differential metabolomics of murine embryos cultured with endometrial cells-conditioned media (CM). Groups of 10, 1-cell murine B6C3F1 × B6D2F1 embryos were cultured in the microfluidic device. To produce CM, mouse uterine epithelial cells were cultured in potassium simplex optimized medium (KSOM) for 24 h. Media samples were collected from devices after 5 days of culture with KSOM (control) and CM, analyzed by reverse phase liquid chromatography and untargeted positive ion mode mass spectrometry analysis. Blastocyst rates were significantly higher (*p* < 0.05) in CM (71.8%) compared to control media (54.6%). We observed significant upregulation of 341 compounds and downregulation of 214 compounds in spent media from CM devices when compared to control. Out of these, 353 compounds were identified showing a significant increased abundance of metabolites involved in key metabolic pathways (e.g., arginine, proline and pyrimidine metabolism) in the CM group, suggesting a beneficial effect of CM on embryo development. The metabolomic study carried out in a microfluidic environment confirms our hypothesis on the potential of uterine epithelial cells to enhance blastocyst development. Further investigations are required to highlight specific pathways involved in embryo development and implantation.

## 1. Introduction

Despite advances in Assisted Reproductive Technologies (ART) [1] and the evolution of in vitro embryo culture systems, the quality of in vitro derived mouse embryos still remains lower than those produced in vivo [2]. The media composition for in vitro embryo culture has been considered a prominent cause for this failure and the effect of the different supplements has been deeply investigated [3,4]. The design of media is complicated, because the components and their concentrations should be determined in order to minimize stress for the cultured embryos and to reflect the concentration of nutrients, electrolyte and macromolecules (e.g., paracrine and autocrine factors, growth factors) present in the lumen of the female reproductive tract (i.e., the oviduct) [5,6,7]. Two possible formulations have been considered: one is based on single embryo culture medium in which the embryo is cultured in a constant medium containing all the supplements needed for its development [8], and the other is based on the so-called sequential culture media designed to mimic the chemical environmental changes that the embryos experiences in vivo moving through the fallopian tube and the uterus [9,10].

The optimal culture media formulation to support in vitro pre-implantation mammalian embryo development has not yet been defined [11]. Currently, one of the most widespread media used for mouse embryo culture is the potassium-supplemented Simplex Optimization Media (KSOM) [12]. KSOM was greatly improved when Ho et al. supplemented the medium with essential and non-essential amino acids, which are present in high concentration in female reproductive tract fluids and are required to maintain cell viability in vitro [13,14]. This allowed to significantly augment embryo development in vitro [15].

To clarify aspects of embryo implantation failure and embryo maternal communication, researchers cultured embryos in contact with a feeder layer of uterus derived cells that directly recreate the natural chemical and physical environment. These studies demonstrated that the co-culture with uterine cells improve in vitro early embryonic development and quality of mouse embryos [16,17,18,19,20]. Cells release diffusible factors, such as bioactive molecules, growth factors and cytokines, that favor embryo development [17], but the molecular mechanisms involved in this embryo-maternal communication still remain not completely understood [18]. Other works suggest that cell-to-cell contact between cells and embryos is not required and that the use of conditioned media may be a sufficient, and more practical alternative to co-culture. Conditioned media culture systems can pose several advantages over co-culture models, such as the absence of foreign cells and the presence of embryotrophic factors [19]. Although several studies demonstrated that the use of oviductal epithelial cells-conditioned media support the development of early embryos in cattle [20,21,22,23], fewer works have been conducted in mouse. For instance, Lee et al. observed no significant changes in blastulation rate when a porous insert was placed between Vero cells and mouse embryos (50%) or when embryos were cultured directly on the cellular monolayer (57%) [24]. However, these values were significantly higher than that of the control (29%), revealing the beneficial effects of cells-conditioned media on embryo culture and viability.

In this work, we evidence alterations of metabolomic profiles of fully developed blastocysts cultured in a controlled microfluidic device using culture medium previously exposed to endometrial cells for 24 h. Metabolomics analysis was carried out by untargeted mass spectrometry (MS) and revealed the presence of metabolites involved in key metabolic pathways which can be associated with improved culture conditions in conditioned media.

## 2. Materials and Methods

### 2.1. Microfluidic Device Fabrication and Preparation for Culture

Microfluidic devices were fabricated in polydimethylsiloxane (PDMS, Sylgard^®^ 184, Dow Corning, Midland, MI, USA). The design of the device has been previously described [25]. Briefly, each device is composed of two PDMS layers fabricated using a standard soft lithographic process using SU-8 photoresist and bonded together using oxygen-plasma. The assembled devices were immediately filled with embryo-tested sterile-filtered water (Sigma Aldrich, St. Louis, MI, USA) and stored at 4 °C until used to preserve hydrophilicity. Before use, devices were sterilized by exposure to UV light (254 nm wavelength for 30 min). The microfluidic device was placed inside a 60 mm MEA tested ART culture dish (Nunclon^®^, Scientific Lab Supplies, Nottingham, UK) and surrounded with 4 mL embryo-tested sterile-filtered water, in order to preserve the humidity inside the dish and to avoid evaporation of the media during the culture. Devices were prepared by adding a 10 μL KSOM (with amino acids, D-glucose and phenol red, Millipore, Watford, UK) medium drop to the channel inlet and drawing the media from the channel outlet, repeating this procedure ten times. 10 μL drops of fresh medium were then added to inlet and outlet before pre-equilibration overnight at 37 °C under 5% CO_2_, 5% O_2_ in a humidified nitrogen atmosphere.

Thawed embryos were cultured in group of ten in the microfluidic device. Before embryo loading, the 10 μL media drops were removed from inlet and outlet ports of the device. Embryos were then loaded through the inlet port by using an EZ-grip embryo handling pipette with a 145 μm diameter tip, which is traditionally used for embryo culture and has a tip size compatible with the inlet port. Medium was then drawn through from the channel outlet port until all embryos entered the central chamber. Next, 20 μL drops of pre-equilibrated KSOM were then added to channel inlet and outlet ports before culture at 37 °C under 5% CO_2_, 5% O_2_ in a humidified nitrogen atmosphere in a benchtop MINC™ Mini incubator (Cook Medical, Brisbane, Australia).

Overall, 39 devices were used in the blastocyst development study and 18 devices for the mass spectrometry analysis. 

### 2.2. In Vitro Embryo Culture in Microdrops

In this study, commercially available, frozen one-cell murine embryos, in vivo derived from the B6C3F1 × B6D2F1 strain, were purchased from Embryo-Tech (Haverhill, MA, USA). According to the manufacturer’s protocol, straws containing frozen murine presumptive zygotes were exposed to room temperature for 3 min and plunged into a 37 °C water bath for 1 min. The straws were cut at the seal and the plug bisected before pushing the contents into a 40 μL drop of M2 medium (Merck Millipore, Watford, UK) in a 60 mm IVF hydrophobic culture dish. Finally, embryos were rinsed twice in pre-warmed KSOM medium and allowed to rehydrate for 10 min in the final rinse droplet at 37 °C under 5% CO_2_, 5% O_2_ in a humidified nitrogen atmosphere in a benchtop MINC™ incubator. Finally, 10 embryos were cultured in 40 µL culture microdrops in 35 mm hydrophobic IVF certified dishes (Nunc™, Thermo Fisher Scientific, Warrington, UK), covered with 5 mL of BioUltra mineral oil (Sigma Aldrich). A total of 360 embryos were used for this study. 

### 2.3. Mouse Uterine Epithelial Cells Culture

Mouse uterine epithelial cells (MUECs) were purchased from Creative Bioarray (CSC-C9063J, New York, NY, USA) and cultured in SuperCult^®^ Complete Epithelial Cell Medium (ECBM, Creative Bioarray) containing 2% fetal bovine serum (FBS, Creative Bioarray, USA), 1% L-glutamine (Creative Bioarray) and 1% antibiotic-antimitotic solution (Creative Bioarray). Cells were maintained in a humidified 5% CO_2_ environment at 37 °C. Trypsinized cells were seeded at a density of 25 × 10^3^ cells/cm^2^ on a 12 well plate coated with 0.1% gelatin solution (EmbryoMax^®^, Merck Millipore). Culture medium was replenished every 2 days until cells reached 70% confluence.

To prepare conditioned medium, ECBM medium was replaced with fresh KSOM when MUECs were 70% confluent. To ensure complete removal of traces of ECBM and FBS, cells were washed twice in Dulbecco Phosphate’s Buffered Saline (DPBS, Sigma Aldrich). Cells were then incubated for 24 h in a humidified 5% CO_2_ environment at 37 °C. Conditioned medium was sterilized by passage in a 0.2 µm-pore-diameter Millipore filter and immediately used.

### 2.4. Embryo Culture in Uterine Epithelial Cells-Conditioned Media

Uterine epithelial cells-conditioned media (CM) was used to culture mouse embryos in microdrops or in devices and results were compared with embryos cultured in KSOM (control). For drop culture, embryos were cultured in oil-covered 40 µL drops (4 µL/embryo) in either control KSOM or KSOM CM, for 5 days or until the developmental stage of fully expanded blastocyst was reached. At the end of culture, spent media samples were collected and immediately frozen at −80 °C for MS analysis. Each experiment was performed in triplicate and replicated 3 times (*n* = 9 drops, 90 embryos for each experimental group).

For device culture, ten embryos were loaded in each device and cultured in either control KSOM or KSOM CM for 5 days or until the developmental stage of fully expanded blastocyst was reached. Each experiment was performed in triplicate and replicated five times (*n* = 15 devices, 150 embryos for each experimental group). At the end of the culture spent media samples were collected from inlet and outlet ports from each device and immediately frozen at −80 °C for MS analysis.

### 2.5. Global Untargeted Metabolomics 

To assess the effects of culture in CM on pre-implantation embryo metabolomics, MS analysis was performed on media samples collected from microfluidic devices or microdrops (control) after embryo culture. Specifically, the same embryo-to-volume ratio has been maintained across the different culture conditions. Embryos were cultured from 1-cell to the expanded blastocyst stage. In this study, overall, the blastocyst stage was achieved at day 4 in microdrops and day 5 in devices. The slightly altered incubation time can be explained by the fact that embryo development in the device microenvironment is partially slowed down and embryos develop to blastocysts stage between 12 and 24 h later than in microdrop culture. However, as previously shown [25], no significant reduction of blastocyst rate or altered embryo metabolism were observed in devices, compared to control microdrops. Spent media collected from microdrops or devices without embryos after the same incubation time used for embryo culture was used as a control. 

At the end of the culture, medium was aspired from each microdrop and from each device and kept at −80 °C prior to sample preparation for analysis. Each sample replica was obtained by pooling spent media collected from 3 different devices (40 µL from each device, for a total of approx. 120 µL). For analysis, 100 µL of sample (from the approx. 120 µL) was used to guarantee consistency between the different biological replicas. Samples (100 µL) were thawed on ice following a standard operating procedure: 300 µL of dry ice cooled methanol was added to individual culture medium samples and incubated overnight at −80 °C. Each sample was then spun down to remove proteins and to aspirate the supernatant for analysis. Reverse-phase liquid chromatography was used, connected to a Thermo Scientific Q Exactive HF (LC-Hybrid Quadrupole-Orbitrap MS/MS) instrument using positive ion mode MS [26,27,28]. The data were thus imported, processed and normalized in Progenesis QI v.2.1 (Non-linear Dynamics, Newcastle, UK). A relative quantitation of the processed data was then completed. All the samples and derived raw data of this study are available on Metabolomics Workbench [29]. As final step, data were processed by a tentative and putative annotations [30] based on accurate mass measurements (error < 5 ppm), isotope distribution similarity, and manual assessment of fragmentation spectrum matching from different databases (i.e., the Human Metabolome Database [31], Metlin [32], MassBank [33], National Institute of Standards and Technology database [34]). Furthermore, spectral match and RT consistencies between experimental data and chemical standards were assessed manually and using a curated in-house library that allowed an increased confidence in the annotation of many features.

### 2.6. Statistical Analysis 

Data were analysed using GraphPad Prism 8 software (Graph Pad Software Inc., San Diego, CA, USA). All data sets were first tested for fit to the normal distribution by D’Agostino-Pearson test for normality. All normal data sets were compared by Student’s t-test, while all non-parametric data were compared by Mann-Whitney U test. In all instances, significance was determined as *p* < 0.05. For metabolic activity experiments, results were checked for statistical differences between groups by ANOVA with post-hoc Bonferroni test.

For MS, compounds with <20% coefficient of variance (%CV) were retained for further analysis. Within Progenesis QI, a one-way analysis of variance (ANOVA) test was used to assess significance between groups and returned a *p*-value for each feature (retention time_*m*/*z* descriptor), with a nominal *p*-value ≤ 0.05 required for significance. Significant features were further filtered using a fold change (FC) threshold ≥ |2| deemed as significant.

## 3. Results

The culture of the embryos was performed in a microfluidic device (described previously in [35]) (Figure 1A). The device presents a central chamber (Figure 1B), with a fluidic volume of 400 nL, where a group of 10 embryos can be loaded and cultured without refreshing the medium to allow blastocyst development [25]. First, the reduction of the volume surrounding the embryos has been shown before to favor embryo development [35,36], without excessive accumulation of depleted metabolites [37]. This is particularly interesting for the analysis completed in this work, where dilution of media composition needs to be avoided. A second advantage of this device is the absence of oil for embryo culture, which facilitates the culture procedure and favors collection and refreshment of medium, as well as collection of embryos and medium sample for analysis.

### 3.1. Blastocyst Development Is Favored by Endometrial Cells

Effect of conditioned media (KSOM CM) on blastocyst development was first assessed by measuring early-cleavage to the 2-cell stage and blastocyst rates of embryos cultured in devices or traditional microdrops. Cleavage rates were measured after 24 h from the thawing as a percentage of the number of early cleaved embryos (2-cells) from the total number of embryos cultured in the drop/device. Blastocyst rates were measured after 96 h from the thawing as a percentage of the number of blastocysts from the total number of embryos in cultured in the drop/device. Results were compared with culture in control media (KSOM). Cleavage rates were comparable for embryos cultured in drops and in devices. In drops, cleavage rates were 96.89 ± 1.59% (*n* = 10) and 95.09 ± 1.64% (*n* = 10) for embryos cultured in conditioned media and control media, respectively. In devices, cleavage rates were 98.7 ± 0.7% (*n* = 23) and 94.9 ± 2.7% (*n* = 21) for embryos cultured in conditioned media and control media, respectively. In terms of blastocyst rate, results were relatively different between drops and microfluidic devices. In drops, the use of conditioned media had no effect on blastocysts development and blastocyst rates were 92.1 ± 2.5% (*n* = 10) for embryos cultured in KSOM and 85.4 ± 3.9% (*n* = 10) for those cultured in conditioned media. In contrast, blastocyst rates were 54.6 ± 6.6% (*n* = 19) and 71.8 ± 4.3% (*n* = 20) for embryos cultured in devices using KSOM or conditioned media, respectively (Figure 2).

Interestingly, cleavage rates were similar for embryo cultured either in microdrops or microfluidic devices in both culture media. We speculate that, for early-cleavage embryo development, the reduction of the culture volume (i.e., increase in biomolecule gradients for compounds present in CM) does not have a significant impact. In contrast, increased biomolecule gradients generated by the reduced culture volume in the microfluidic device, could significantly impact the development of later stage blastocysts.

### 3.2. Exposure to Uterine Cells Alters Metabolomics Profiles of Individual Blastocysts

To assess the impact of culture using mouse uterine epithelial cell-conditioned media, the metabolite composition of spent CM collected from microfluidic devices at day 5 of embryo culture was measured by untargeted LC-MS/MS. Data were compared either to control media, fresh KSOM CM at day 0 (Day 0 CM), or to media incubated in the device for 5 days without embryos (Day 5 CM). As a control, results were compared to those obtained using KSOM. Figure 3 shows the overall LC-MS/MS results of the different experimental sample groups. A total of 2338 compounds were detected from the analysis (CV < 20%). From a statistical evaluation by principal component analysis, a clearly different distribution of metabolic profiles was observed among all the analyzed experimental groups (Figure 3A). This confirmed also the quality of the analysis, since all the experimental groups are clearly separated and characterized by an altered levels of metabolites. Those differences are also visible in the heatmap visualization shown in Figure 3B where colors are displayed by relative abundance and all metabolic compounds are represented as rows. From the heatmap, is possible to identify three main clusters of metabolites. The first cluster from the top is likely to be associated with metabolites taken up by embryos, as it includes features which are significantly reduced in spent media from embryo culture. The second cluster includes species differentially expressed in all the experimental groups, thus difficult to associate to any specific condition. Finally, the third cluster shows compounds significantly altered in spent media collected from devices compared to control, representing metabolites associated to embryo culture and to the microfluidic environment.

### 3.3. Differential Expression of Metabolites in Spent CM Samples Compared to KSOM

To further investigate key metabolic markers that can be directly associated to the use of conditioned media, the three pairwise comparisons were compared, as summarized in Table 1.

From the first comparison, 621 features were significantly (*p* < 0.05, FC ≥ |2|) altered in embryo culture CM compared to embryo culture KSOM, meaning that the metabolite composition of the medium is significantly different once the embryos reach blastocyst stage. Of these 621 compounds, 385 illustrated an up-regulated trend and 236 a down-regulated trend in the CM group compared with the control KSOM group (Figure 4A). Interestingly, the volcano plot shown in Figure 4A spotlights the dysregulated compounds detected only in one of the groups, represented by the metabolites on the extreme right and left of the graph. These include 7 compounds unique to spent embryo culture KSOM and 61 to spent CM from embryo culture. The 61 species are metabolites produced by embryos as a consequence of culture in CM or compounds already present in CM due to the conditioning of the media with uterine epithelial cells, including N-gamma-L-glutamyl-D-alanine, N,N’-bis(4-nitrophenyl)-urea, 6-methyltetrahydropterin, glycerophosphocholine, glutaminylleucine, 1-methyladenosine, uracil, 3-dehydroxycarnitine, 7-methylguanine, inosine diphosphate, thiamine, guanine, L-alpha-aspartyl-L-hydroxyproline and 5-phosphoribosylamine.

From the comparison of CM and KSOM media after 5 days (Figure 4B) in absence of embryos, among the 369 illustrated an up-regulated trend in CM, 84 were only found in the CM group and thus can represent compounds specifically released by uterine epithelial-cells into the medium. In contrast, nine features were unique to the KSOM group and, as such are purely related to the not-conditioned KSOM medium. Furthermore, from the comparison between the CM and the KSOM medium at day 0, it was possible to identify the compounds up-regulated (491) and down-regulated (76) in CM, as well as features uniquely expressed in the CM (113) and in KSOM at (3) at the beginning of the embryo culture (Figure 4C).

### 3.4. Pathways Overrepresentation Analysis Evidenced Alteration of Key Metabolic Pathways in CM 

From the comparison between day 5 embryo culture CM and day 5 embryo culture KSOM groups, a total of 353 significant (*p* ≤ 0.05, fold change ≥ |2|) compounds were putatively identified using available libraries. This allowed to obtain a list of up-regulated (208 species) and down-regulated (145 species) metabolites in spent CM compared to control KSOM. The attributed metabolites were combined and used for pathway overrepresentation analysis. Potential biomarkers were mapped using the Mus musculus (mouse) KEGG pathway library in Metaboanalyst 4.0, a web-based program for pathway enrichment and topology analysis. Figure 5 presents a list of the matched overrepresented pathways for compounds significantly increased (Figure 5A) and decreased (Figure 5B) in day 5 embryo culture CM, compared to day 5 embryo culture KSOM.

As summarized in Figure 5A, the most affected metabolic pathway for the compounds significantly increased in CM was arginine and proline metabolism, with increased abundance of L-glutamate, L-proline, creatine, S-adenosyl-L-methionine and hydroxyproline. The metabolic pathway of the amino acids alanine, aspartate and glutamate was also significantly affected, with the increased abundance of the metabolites L-alanine, L-glutamate, succinate and 5-phosphoribosylamine.

An increased production of the metabolites, uridine, thymine, deoxycytidine, deoxyuridine, uracil, guanine, xanthine, inosine diphosphate, deoxyinosine diphosphate, deoxyadenosine, hypoxanthine and 5-phosphoribosylamine, from embryos cultured in CM indicates an effect on DNA bases (purine and pyrimidine) metabolism pathways which are critical during embryo growth.

On the other hand, looking at the list of compounds significantly decreased in the embryo culture CM at blastocyst stage (Figure 5B), it was possible to observe a reduction in abundance of metabolites L-glutamine, L-aspartate, L-serine, L-tryptophan, and L-tyrosine which implies a potential effect of CM on aminoacyl-tRNA biosynthesis. Other amino acid metabolism pathways were affected, such as phenylamine metabolism and cysteine and methionine metabolism, with the decreased abundance of L-tyrosine, phenylacetylglycine, L-serine, L-cystathionine, L-cystine, L-glutamine and L-aspartate.

Finally, using available libraries we identified hundreds of compounds by pairwise comparisons of media at the end of the culture, at day 5 in absence of embryos and at the beginning of the culture. Figure 6 shows in detail the features that were either up-regulated or down-regulated in the CM groups compared to the KSOM groups. The results show species that are uniquely expressed in the pairwise comparisons (respectively 53, 61 and 86), as well as 182 features that were common between the 6 groups regardless of incubation time and embryo culture.

Metabolites produced or consumed by embryos as a result of culture in CM in the device can be directly found from the analysis of the 53 species arising from the comparison between the two spent embryo culture media. In detail, the expression of 24 metabolites was at least 2-fold higher (*p* < 0.5) in CM compared to KSOM including 5-phosphoribosylamine, S-adenosylmethionine and 5-hydroxy-L-trypthophan, which are involved in the metabolism pathways of purine and several amino acids (cysteine, methionine, alanine, arginine, aspartate, glutamate, proline, tryptophan). On the other hand, the 29 metabolites that were at least 2-fold higher abundant in KSOM compared to the CM, included glutamylmethionine, glutamyltyrosine, 9-methylxanthine and dodecanoic acid. None of these compounds were found to be involved in metabolic pathways characteristic of mouse embryo development.

The conditioned medium presents already at the beginning of the culture 79 compounds with at least 2-fold increase abundance compared to pure KSOM. These includes 5-oxoproline, spermidine, L-glutamine and deoxyribose, which are intermediate of some metabolic processes such as glutathione metabolism, glutamine and glutamate metabolism, nitrogen metabolism and arginine metabolism. These metabolites are likely to be biomolecules produced by epithelial cells and released into the KSOM during the conditioning of the media. Their increased abundance into the culture media can have an impact on in vitro pre-implantation embryo development. Obviously, the conditioned medium also presents a lower abundance of some metabolites, including alanyl-Isoleucine, phenylalanyl-arginine, N-acetylisoleucine, arginyl-isoleucine, *m*-methylhippuric acid, 1*H*-indole-3-acetamide and 2-diethylaminoethanol, as result of the previous consumption by the uterine epithelial cells.

## 4. Discussion

In this work we assessed the possibility to improve the culture conditions by using mouse uterine epithelial cells-conditioned media for the culture of murine embryos from 1-cell to blastocyst stage. Increased blastulation rates were observed in blastocysts cultured in cells-conditioned media when compared with control.

Interestingly we did not observe any significant difference in the cleavage rate in both the microdrops and microfluidic devices either cultured in KSOM or in uterine epithelial cells conditioned medium. This effect can be related to the reduced culture volume present in the microfluidic device, which generates an increased biomolecule gradient in the surrounding embryo microenvironment. This could have a more significant impact of the use of CM on the development of later stage blastocyst (expanded, hatching) compared to early-stage embryos (2-cell). Early cleavage stages and blastocyst embryos require in fact the availability of different sets of nutrients and show different metabolic activities [38,39]. A further study will include a more specific analysis of the medium composition at different developmental stages, to assess the specific metabolomic profiles of the embryos during development and correlate these with the cleavage rate.

The level of blastocyst rates observed in microfluidic devices is lower than the one in the control microdrops. This effect was not observed in previous studies, where the blastocyst rate exceeded 80% and was comparable to the traditional culture [35]. This difference can be ascribed to the specific mouse strain, the different batches of media and the different operator skills. These values are within the intrinsic variability observed in literature for mouse embryo blastocyst rate [40,41]. These values are however still comparable [36] or higher [42] than the ones achieved with other comparable microfluidic systems. This might be associated with the reduced culture volume and optimized biomolecule gradients provided by the device microenvironment which better resembles that found in vivo. In particular, our results showed that in devices the use of cells-conditioned media, significantly (*p* = 0.037) improved blastocyst development compared to control KSOM. Biomolecules and growth factors produced by cells and present in the conditioned media might have a greater effect on embryos cultured in such reduced volume, rather than in bigger culture drops, because of reduced embryo-to-media ratio and optimized gradients.

Importantly, LC-MS/MS analysis of the culture medium allowed the identification of important biological markers revealing insights into embryo metabolic activity. For this analysis, equal media volumes from the two culture platforms were considered in order to maintain a constant embryo-to-media ratio. We assumed that the metabolite content measured in culture media drops is representative of that in the culture chamber, since slow diffusion phenomena induce transport of biomolecules along the channels to the media drops. Pathway overrepresentation analysis showed that some key metabolic pathways were enriched for the metabolites consumed by the embryos during culture in microfluidic devices using conditioned media. Among them, metabolic pathways of several amino acids and DNA bases that have a critical role in embryo development were significantly affected. For instance, the alanine, aspartate and glutamate metabolism pathway is of fundamental importance during embryo development. Amino acids serve not only to provide energy but also to maintain embryo function by preventing cellular stress induced by suboptimal culture conditions in vitro [43,44]. Noteworthy, the increased abundance of S-adenosyl methionine might indicate an effect on the de-novo methylation of DNA process, which is essential for early mammalian development and for the critical process of gene imprinting [45]. S-adenosyl methionine is indeed necessary for the activity of DNA methyl transferase activity and its detection could be correlated to an increased activity of, not only arginine and proline metabolism, but also DNA methylation in blastocysts cultured in the device [46]. Similarly, pyrimidine metabolism pathway is required throughout embryo development as inhibition of de novo pyrimidine nucleotide synthesis can prevent blastocyst development [47]. Purine metabolism is a complex scheme of enzyme reactions which are described as two pathways: the purine salvage pathway and the de-novo synthesis pathway. These metabolic pathways are fundamental for pre-implantation embryo development as they allow synthesis of purine precursors for nucleic acid formation [48]. The detection of two end- products, xanthine and hypoxanthine, confirmed that those two pathways are both active in pre-implantation embryo development. Xanthine has indeed been reported to be the end-product of purine degradation in the mouse pre-implantation embryo development, while hypoxanthine is an activator of purine salvage pathway [49,50].

The up-regulation of compounds involved in those metabolic pathways can be explained as higher metabolic activity of blastocysts cultured in CM, and this could reveal a beneficial effect on pre-implantation embryo development. In contrast, another hypothesis might consider that blastocyst cultured in CM, compared to KSOM, and instead consumed those metabolites by a reduced amount. Despite being difficult to elucidate specific function and activity of these pathways, it is clear that the use of CM has a significant impact on embryo metabolomics and these data need to be investigated further. In contrast, the down-regulated compounds could be associated to metabolic pathways whose activity is reduced in blastocyst cultured in CM.

Overall, this comprehensive metabolomics analysis confirms that the uterine cells change the metabolism of the embryos through the duration of the culture.

Noteworthily, those results indicate an effect of conditioned media in protein and DNA synthesis, fundamental process during embryo development, which may drive to higher cell number and improvement in pre-implantation embryo development.

Overall, those findings reveal the feasibility of performing metabolomics analyses of blastocysts from small volumes of samples collected from the microfluidic device. It is now known that “omics” techniques can provide new means to monitor embryo development and assess embryo quality in a more comprehensive way than morphokinetics alone. Thus, this set of data represents an additional proof that the future integration of such techniques with miniaturized and in vivo-like microfluidic culture systems has the potential to help the progress in ART techniques and IVF outcomes.

This microfluidic technology presents innovative aspects compared to traditional embryo culture platforms and currently available microfluidic platforms. First, this system eliminates the need of potentially embryo-toxic mineral oil, thus reducing the sampling of medium from the culture, and avoiding contamination of the samples that can alter the embryo development, impair the sample preparation for IMMS or introduce unneeded compounds.

This study represents the first analysis of the crosstalk between the uterine environment and the embryo development by means of untargeted metabolomic investigation of murine embryos cultured in a microfluidic devices. However, it is difficult to ascribe any of the evidenced metabolic effects to neither the presence of uterine cells nor the specific embryo-uterine crosstalk in vivo [51]. The presented results confirm the flexibility of the platforms for non-invasive embryo quality assessment. The clear and consistent distinction between sample groups suggests that that the method and the culture conditions are stable and support and limit deviations of the results.

Further investigations are required to identify and correlate the specific identified compounds in the media composition with in vitro embryo development and the pregnancy outcomes after embryo transfer.

## Figures and Tables

**Figure 1 cells-10-01194-f001:**
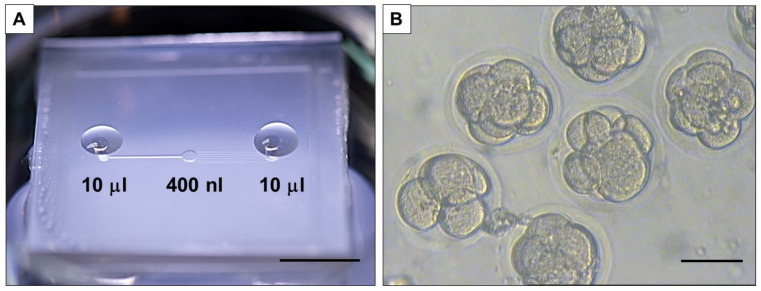
Microfluidic device. (**A**) The device is placed in a 60 mm petri dish for loading medium and kept in the incubator for the duration of the culture (scale bar 1 cm). (**B**) The developing embryos visualized in the central chamber in brightfield (scale bar 100 µm).

**Figure 2 cells-10-01194-f002:**
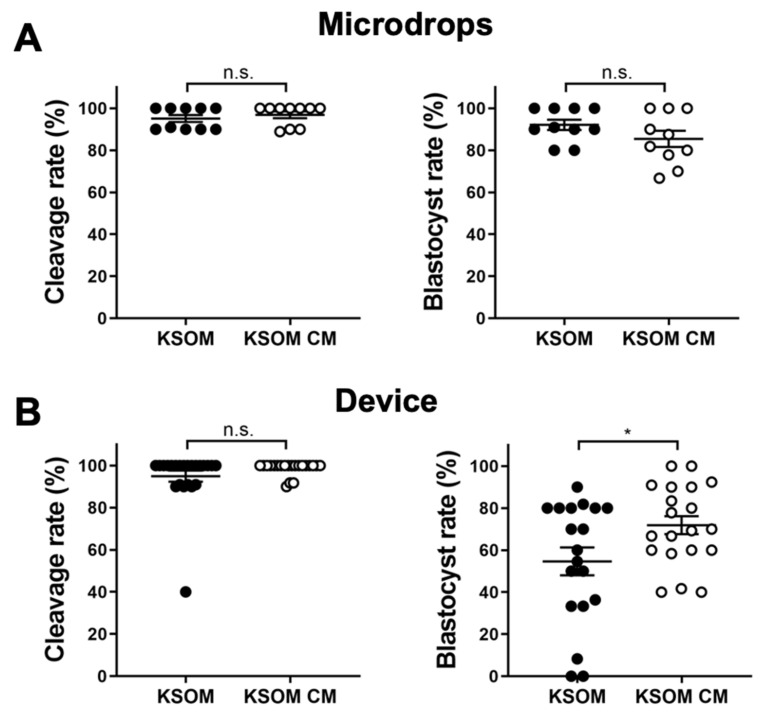
Effects of CM on embryo development. Cleavage rates (left) and blastocyst rates (right) of embryos cultured in drops (**A**) or in microfluidic devices (**B**). Each dot represents an individual culture, in which 10 embryos are cultured. Data presented as mean ± SEM. Non significant (n.s.) differences and significant (*) comparison are shown (* *p* < 0.05).

**Figure 3 cells-10-01194-f003:**
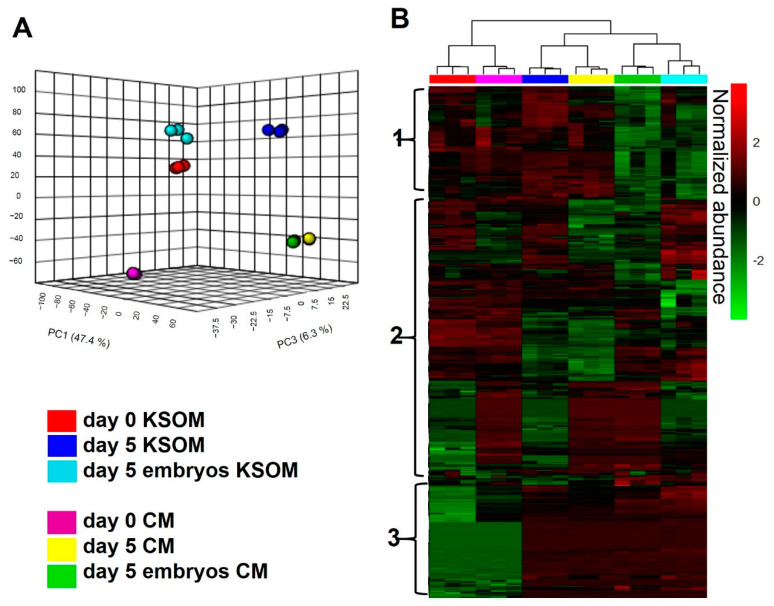
The metabolite composition of the media used for embryo culture is considerably changed by uterine epithelial cells. LC-MS/MS global metabolomic profile analysis of blastocysts cultured in KSOM or CM in devices. (**A**) PCA plot for LC-MS/MS data of medium samples collected from devices in different experimental conditions (*n* = 3 for each group): spent KSOM (blue) or CM (yellow) from devices without embryos, spent KSOM (cyan) or CM (green) from devices with embryos and the relative controls. (**B**) Heatmap representing LC-MS/MS data. Compounds are presented as rows and sample replicates as columns and processed by using Euclidean distance and Ward clustering via Metaboanalyst 4.0 for Pareto scaled, log transformed, and averaged group data. Colours are displayed by normalized abundance, ranging from low (green) to high (red).

**Figure 4 cells-10-01194-f004:**
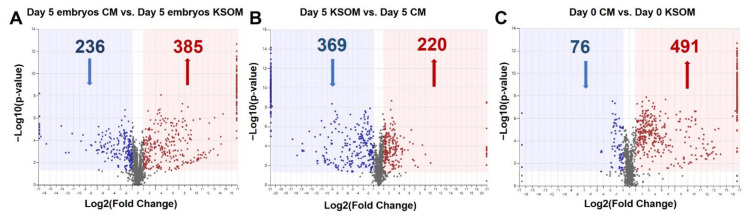
Volcano plot showing the distribution of metabolic compounds for the pairwise comparison. Pairwise comparison of (**A**) day 5 embryo culture CM vs. day 5 embryo culture KSOM, (**B**) Day 5 CM vs. Day 5 KSOM, and (**C**) Day 0 CM vs. Day 0 KSOM. Significance criteria: *p* ≤ 0.05, fold change ≥ |2|. Volcano plots combine the *p*-value measured by ANOVA, expressed as −log_10_(*p*-value), with the magnitude of the change in relative abundance, expressed as log_2_(fold change), between the groups considered in a particular pairwise comparison.

**Figure 5 cells-10-01194-f005:**
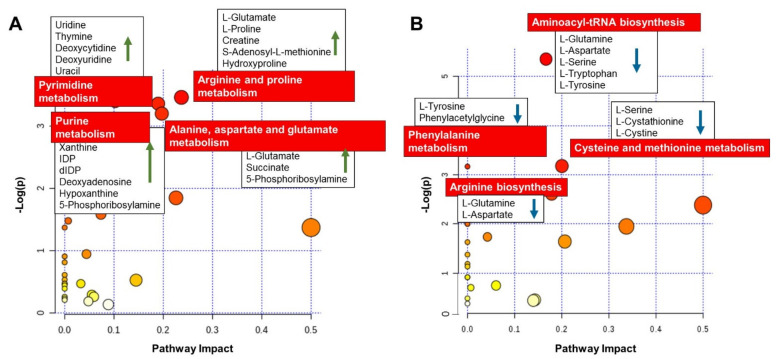
Metabolic pathways of significantly changed metabolites between day 5 embryo culture CM and day 5 embryo culture KSOM. (**A**) Increased metabolites uniquely produced by embryos when exposed to uterine epithelial cells. The graph presents a list of the matched overrepresented pathways for compounds significantly increased in day 5 embryo culture CM, compared to day 5 embryo culture KSOM, arranged by *p*-values on Y-axis, and pathway impact values on X-axis. The enriched pathways for the 208 increased metabolites identified in the CM group are represented by nodes of different size and color gradient corresponding to the significance of the pathway ranked by impact score (the larger the circle the higher the impact score) and *p*-value (yellow: higher *p*-values and red: lower *p*-values), respectively. The red labels highlight the main pathways. Five of these pathways had a *p*-value < 0.1 and 16 had pathway impact value higher than 0, which are the cut-off values for relevance. (**B**) Decreased metabolites uniquely produced by embryos when exposed to uterine epithelial cells. Only pathways with *p*-value < 0.1 were considered significant. Of these pathways, 10 had impact value higher than 0 and 10 had a *p* value < 0.2. The most significantly affected (*p* < 0.1) metabolic pathways are labelled and represented with larger red dots.

**Figure 6 cells-10-01194-f006:**
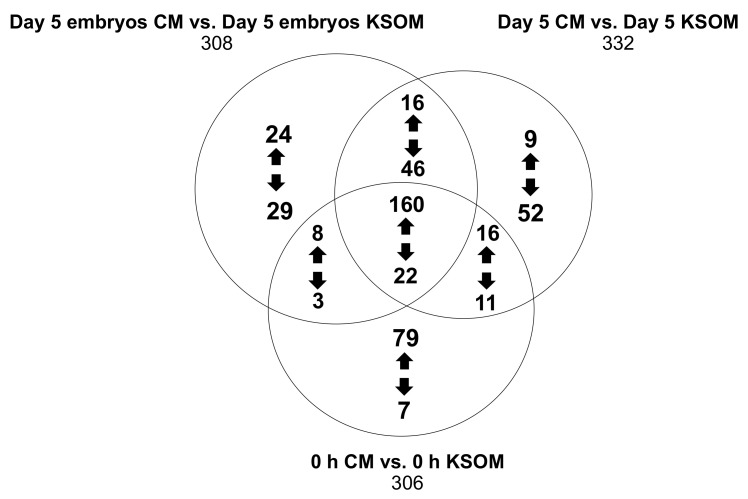
Venn diagram showing the distribution of the number of metabolites between the three pairwise comparisons.

**Table 1 cells-10-01194-t001:** Overview of differentially abundant metabolites in the pairwise comparisons, day 5 embryo culture CM vs. day 5 embryo culture KSOM, day 5 CM vs. day 5 KSOM, and day 0 CM vs. day 0 KSOM. Spent media collected from devices in absence of embryos and fresh media at day 0 were considered as controls.

Pairwise Comparison	Cond A	Cond B	# of Significant Compounds *p* ≤ 0.05 and FC ≥ |2|
n.1	Day 5 embryos CM	Day 5 embryos KSOM	621
n.2	Day 5 CM	Day 5 KSOM	567
n.3	Day 0 CM	Day 0 KSOM	589

## Data Availability

Raw data supporting the reported results can be found in this database https://doi.org/10.21228/M8W99G.

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
