# Peer review of "Metabolomic Analysis Evidences That Uterine Epithelial Cells Enhance Blastocyst Development in a Microfluidic Device"

_cells, 2021, doi:10.3390/cells10051194_

Round 1
Reviewer 1 Report
Reviewer Comments:
The paper submitted by the Authors investigated the “Metabolomic analysis evidences that uterine epithelial cells enhanced blastocyst development in a microfluidic device”. The experiments were well prepared. However, the improvements are required throughout the manuscripts to improve the overall quality of the manuscript and some concerns need to be explain must.
- Line # 214: “blastocyst rates observed in microfluidic devices is lower” and line # 221: cells-conditioned media, significantly (p=0.037) improved blastocyst production 221 compared to control KSOM”, must provide the explanation in which terms authors says that “blastocyst rate” and “blastocyst production” and what is difference how it is lower in the microfluidic device and how it improved. The paragraph is very ambiguous.
- There is no difference in the cleavage rate in both the microdrops and microfluidic devices either cultured in KSOM or in uterine epithelial cells based CM, author must explain the reason in the result section briefly about the requirement and difference of gradients to support the early stages and later stages of embryonic development.
- Table #1 is missing and also there is a repetition of sentences at line # 263 and 268.
- Discussion section needs a detailed elaboration and explain of how the culture of embryos in the microfluidic device improve the development of embryos, effect of embryo to media ratio and optimization of gradients with references.
There are some minor comments also:
- The heading of result 4 and 5 is missing. Does the figure # 4 and 5 included under the result heading 3.2? It is better to explain the result of figure 4 and 5 under separate result heading. It gives more clarity to the readers.
- Line # 66 include the specific bovine reference article regarding the bovine-oviduct-epithelial-cells and embryo development (Sidrat et al 2020)
- Explain M2 media at line # 103 for washing of embryos.
- In whole materials and method section the units are not properly written, for example μl must be write as μL.
- The chemical formula name for the CO2 and O2 must be corrected by using subscript.
- The total volume unit at line # 183 is missing, also check line # 182 is 400nL or 400 μL.
- Figure 1A is not mentioned in the text. To mentioned the figure in the text author did not use the same format. Must do the correction either short “Fig” or full “Figure” according to the journal’s formatting style.
- Line # 277 figure number is missing.
- Correct the formatting of the references in the text according to the journals formatting style.
Reviewer 2 Report
Comments to the Authors
The manuscript from Mancini et al. investigates the potential of uterine epithelial cells to enhance blastocyst development performing a metabolomic study in a microfluidic environment. The embryos culture in presence of CM in the microfluidic device, modified the metabolism of embryos and increased the rate of blastocysts compared with the embryos cultured in the device but in absence of CM. However, overall this work is poorly conducted and fails to provide strong evidence to support the authors conclusions. Also, some data (commented below) are not straightforward and the technical concerns remain.
Abstract: The blastocysts rates indicated in this chapter are not corresponded to that indicate in the results.
Materials and Methods: Are the embryos used in this study obtained in vivo or in vitro? Why do the authors use the frozen and not fresh embryos?
When the authors calculate the cleavage rate, it is not clearly explained, after how many hours (calculated from the thawing/insemination/mating?) of embryo culture the rate is calculated.
The authors made a comparison between culture made in drops vs. that made in the device. In my opinion it is difficult to evaluate a beneficial effect of CM without this previous analysis.
The results and comparisons are difficult to follow in the way they are written. It is recommended to rewrite the materials and methods with special attention to experimental design to be able to understand the results and confirm the conclusions.
The discussion is not a discussion; no information from literature is used to compare the results.
Line 105: the authors say that the embryos are cultured in 1 ul/embryo microdrops but, then, white that culture was made in 4 ul/embryos drops (line 125). Which of the two sentences is correct? The authors did not detail the atmosphere conditions of embryos culture.
Line 118: the cells are incubated at 37 °C for 24h without any specific atmosphere conditions?
Line 126: The authors said that the embryos were cultured for 5 days but in line 143 said that the blastocysts stage was achieved on day 4. Which of the two sentences is correct? If the embryos reach the blastocyst stage in the drops after 4 days of culture, how do the authors justify the delay of embryos cultured in the device?
Line 149: the authors say that the samples volume is 100 ul but before (line 144) they write 40 ul
Lines 167-177: What test the authors used when comparing the embryo developmental rates? If I well understand, the authors compared cleavage or blastocyst rate in CM compared to KSOM but in the Statistical analysis section the comparison between two groups is not mentioned. Did all datasets pass the test of normal distribution? Based on Figure 2 the data appear to have a no parametric distribution.
Lines 182-185: If I well understand, the volume of the central chamber in which 10 embryos are located contained 400 nl of medium. Is this volume enough to guarantee a good quality growth of embryos? What means that the total volume is 20,58 L? Overall, this sentence does not make sense.
Figure 1: I’m confused on the volume that the authors used for the embryos culture in the microfluidic device.
Lines 200-208: The authors said that the total number of embryos for the study was 180. Why the authors calculated the rate of cleavage and blastocyst only in a small number of embryos. Based on the plot, I have some doubts about the normality distribution of the date.
Figure 2: The names of the groups are different from the previous description. The sample names should be align in the text
I didn’t understand if the authors compared each group to each other or made a comparison two by two (Day 0 CM vs. Day 0 KSOM; Day 5 CM vs. Day 5 KSOM; Day 5 embryo CM vs. Day 5 embryo KSOM)
Table 1 is lacking
Line 277: the number of figure is lacking
How is it possible to determine the number of unique compounds from the figure 4?
Line 306: the total of 353 significant compounds were isolated from the 621 features obtained after the first analysis?
The Figure 5 and its legend are not clear. What do the red squares represent?
Line 277: the number of figure is lacking
Figure 6: I didn’t understand how the authors arrived at defining the Venn diagram shown in Figure 6. This figure is not mentioned in the text.
The authors should eliminate all comments or discussion from the Results chapter.
Reviewer 3 Report
Mancini et al. present data on the effects of embryo culture medium that had been conditioned by uterine epithelial cells on the development and metabolomic activity of murine embryos. The data is novel and interesting, but there are several confounding factors with the experimental design that make interpretation of that data difficult. First, as stated by the authors in lines 224-227, the uterine cells likely secreted growth factors and cytokines into the conditioned medium that could affect the embryos. Therefore, there is no way to know if the observed differences in development were caused by the altered nutrient composition of the medium or the presence of the growth factors. Second, all of the metabolomic data was generated from embryos cultured in the microfluidic devices, but embryonic development was compromised in this environment. Embryonic development was both reduced (lower overall blastocysts production) and delayed (day 5 vs day 4). This suggests that the culture environment was flawed and resulting metabolomic data may reflect stressed, non-viable embryos. Finally, the density of embryos was different between drops (10 embryos in 40 ul) and the devices (10 embryos in 400 nl or a total of 20.58 ul). Why weren't 20-21 ul drops used for the drop culture and why wasn't this medium collected for metabolomic analysis. The manuscript could be improved if the authors discuss these factors in the Discussion. Other comments:
1) Lines 86 and 117: Please indicate what protein was present in the KSOM.
2) Line 91 and 105: Please indicate the number of embryos and the total culture volume here, rather than lines 179-190.
3) Line 117: Please describe the process for removing all traces of ECBM and FCS from the uterine cell culture before replacing with KSOM. Were the cells rinsed several times with KSOM? Could the conditioned medium be contaminated with ECBM and FCS?
4) Lines 126 and 131: Culture duration was described as, "...day 5 or until; the developmental stage of fully expanded blastocyst was reached." What does this mean? Were embryos cultured beyond day 5 for some replicates?
5) Lines 143-144: Blastocyst development was delayed by 1 day in devices? This should be discussed.
6) Line 182-183: Is there any data available on the diffusion of nutrients through the small channel? Is it possible that 10 embryos depleted some nutrients in the 400 nl of medium immediately surrounding them and these nutrients were no replenished by diffusion from the wells at the ends of the channel? I assume the entire volume (20.58 ul) was pooled for analysis?
7) Line 183: Should this read, "First, the reduction..."
8) Line 178: The Results section is very long and a bit confusing. A considerable amount of this section would actually be better suited to the Discussion.
9) Lines 179-189: This paragraph does not contain any Results.
10) Lines 214-227: This paragraph does not contain any Results. If the authors are suggesting a differential effect of conditioned medium based on the strain of the mouse, this should be moved to the Discussion. They should also consider that culture in the device stressed the embryos, which made them more dependent on the growth factors in the conditioned medium for development?
11) Lines 261-273: It appears a Table is missing?
12) Lines 442-447: The results do not indicate that any of the described effects are specific to uterine cells or have any relevance to embryo-uterine crosstalk in vivo. The authors should refer to Edwards et al., 1997 (Mol Reprod Dev 46:146-154) who showed that some changes resulting from co-culture are cell type specific and others are consistent across cell types.
Reviewer 4 Report
Interesting study, I have only few minor comments.
- I couldn't find Table 1?
- you already discussed what could be the reason for lower blastocyst rat in microfluidic device, but I am wondering what is the exchange of gases in such device? could this be the reason for lower blastocyst rate?
Author Response
Interesting study, I have only few minor comments.
- I couldn't find Table 1?
We have now inserted the Table in the revised version.
- you already discussed what could be the reason for lower blastocyst rat in microfluidic device, but I am wondering what is the exchange of gases in such device? Could this be the reason for lower blastocyst rate?
The gas permeability of the PDMS is definitely different from the gas permeability in the oil covering the drops in drop culture. On this, we are currently working on measuring the oxygen gradient in the device during the culture to assess any changes in the metabolic activity of the embryos. We have however carried out several cultures with this device and the blastocyst rate is in average >80%, and the gas permeability does not seem to significantly alter pH, osmolality and evaporation of the medium during culture (data not shown).
Round 2
Reviewer 2 Report
This is the revised version of previously submitted manuscript titled: metabolomic analysis evidences that uterine epithelial cells enhance blastocyst development in a microfluidic device.
The authors have addressed concerns and revised the manuscript adequately. The controversies in methodology have been clarified and the presentation has been essentially improved.
This paper presents findings in a topical area of identification of platforms for non-invasive embryo quality assessment, which makes it interesting for the research community as well as for clinical practice and warrants publication in Cells.
Minor revision
Table 1 is repeated two times
Figure 4: The authors should uniform the Figure 4B. In the figure the pairwise comparison is between KSOM vs. CM whereas in the legend the authors write CM vs. KSOM.
Author Response
Thanks for your feedback, we now fixed the Table and the references to Figure 4.
Reviewer 3 Report
The authors have adequately addressed my concerns on the original, submitted manuscript.
Author Response
Thanks for your feedback.